# Smart Cities and Data Analytics for Intelligent Transportation Systems: An Analytical Model for Scheduling Phases and Traffic Lights at Signalized Intersections

**Fatih Gunes** [1,*] **, Selim Bayrakli** [2] **and Abdul Halim Zaim** [1]

[1] Department of Computer Science, Istanbul Commerce University, Kucukyalı E-5 Kavsagi Inonu Cad. No 4 Kucukyali, Istanbul 34840, Turkey; azaim@ticaret.edu.tr

[2] Department of Computer Engineering, Air Force Academy, National Defence University, Yesilyurt, Istanbul 34149, Turkey; hbayrakli@hho.edu.tr

[*] Correspondence: fatih.gunes@istanbulticaret.edu.tr; Tel.: +90-507-153-60-46

**Abstract:** This paper is intended to improve the performance of signalized intersections, one of the most important systems of traffic control explained within the scope of smart transportation systems. These structures, which have the main role in ensuring the order and flow of traffic, are alternative systems depending on the different methods and techniques used. Determining the operation principles of these systems requires an extremely careful and planned study, considering their important role. Performance outputs obtained from the queue analyses made in previous studies formed the input of this study. The most important techniques are used in the effective control of intersections, such as signal timing: in particular, the use of effective green time and order of the transitions between phases. In this research, a traffic-sensitive signalized intersection control system was designed with the suggested methods against these two problems. The sample intersections were selected from three cities with the highest population density as the case study area. In the analysis of the performance of the connection arms of the selected intersections, flow intensity data were taken into consideration, as well as the arrival and service rates. Based on this, the outputs of the two proposed models regarding the use of phase transitions and effective green durations were compared with two other adaptive control systems and their positive results were shared. The results showed that signalized intersections, operating with a well-planned and correctly chosen technique, better regulate density and queuing.

**Keywords:** smart cities and data analytics; signalized intersections; scheduling traffic lights and phases; dynamic traffic lights assignment; intelligent transportation systems

## 1. Introduction

With the increasing population, it has become inevitable for city planners to integrate technology into management systems and to use it to improve the quality of daily life. The smart city concept, which brings together technology and the management components of cities, has the potential to improve many areas with its widespread systems. Developing communication systems, wireless network technologies, and infrastructure equipment have opened the door to many innovative applications in the city management. It is seen that the technologies developed in the field of transportation, which is one of the topics of the smart city approach, are increasingly used to facilitate urban life.

Research and development in the field of smart city technologies continue to spread and develop. The implementation and development of these technologies that produce better living standards and value are continuously studied and discussed by adapters of the business, such as city planners, architects, and city managers. The pace of change brought about by these approaches has rapidly improved the life and welfare level of people living in urban centers. There are many studies of researchers or city managers working in

this field to develop the smart city concept and improve living standards. Especially as a result of studies on city residents, research is carried out to determine the indicators that make up the smart city concept [1]. The smart city model, which comprises the collaboration of technology and city management processes, affects many environmental factors. This model has begun to be used in many areas, such as health, education, energy, accommodation, infrastructure, agriculture, environment, and especially in transportation.

There are many environmental and economic benefits of reducing traffic congestion and density, such as the reduction in the time wasted in transportation, the reduction in traffic accidents, and increased efficiency of fuel consumption. Transportation systems are important components in terms of social and living standards that can directly improve people's life quality and make a difference in industrial activities in a sectoral sense. Intelligent Transportation Systems, which stand out in this context, are some of the areas where cities invest the most in terms of technology and infrastructure [2].

Applications developed or likely to be improved in the transport context and their possible benefits formed the basic idea of this article study. Traffic management, which has become the most important problem of cities, especially as a result of increasing urbanization and population density shifting from rural areas to cities, came to the fore as the starting point of the study. In this context, many factors are presented as the main aspect of urban traffic management. Studies such as observations, interviews, and literature reviews have concluded that the problem is especially at the intersections of urban connection roads. Intersections are the most basic infrastructure element designed and implemented by city planners over many years to manage traffic at these points [3].

Signalized intersections and traffic lights, which are the main urban traffic regulatory controls, are especially used at the intersections of connection roads coming from more than one direction. The travel times of vehicles in urban traffic can be prolonged due to insufficient traffic light control. These waiting points and queues that may be experienced affect more than one intersection or transition route. Optimum control of traffic lights, using advanced sensors and smart optimization algorithms, is therefore very critical. Optimization of traffic light times increases the effective road capacity and traffic flow and can prevent traffic congestion [4].

Many different approaches to traffic light control and optimization problems have been addressed so far. It is seen that the studies conducted in this context focus on problems such as delay minimization, time optimization, queuing, and estimation. All the research and development in the field of smart city technologies focus on the solutions to these problems. Developments in many areas, from sensors to camera sensors and traffic control processors to SCADA systems, contribute to the spread of traffic control systems [5]. In this sense, this article is intended to examine the city center signalized intersections, to improve their efficiency and to improve them with the suggested methods. As a result of the field studies and interviews with the authorized units, it is observed that many methods and methodologies are tried and applied. The made studies improved studies on issues such as signal planning, phase sequences, and performance measurements of intersections located at important points to handle the traffic flow.

### 1.1. Relevant Studies

Many studies have been carried out to facilitate signaling at intersections. Early work in traffic engineering began with the recommended methods for vehicle delay calculation. Some of the first proposed methods for delay calculations were developed by Webster and Akçelik, and all subsequent research was a continuation of these two studies [6]. Since the study [6], performance improvement studies have begun to calculate signal durations. As a continuation of these methods, studies were carried out and described in the Highway Capacity Manual (HCM) in the literature, extending the delay analysis with research published in different years—1985, 1994, 1997, 2000, and 2010 [7–11]. In the further version of these studies, it is considered that performance improvement and optimization

studies are mainly carried out using optimization methods, linear programming, fuzzy logic, decision-making models, and genetic algorithms.

Hartanti et al. aimed to optimize traffic lights using fuzzy logic to prevent traffic congestion at intersections [12]. Harahap et al. used the queuing theory for modeling and simulating waiting at traffic lights [13]. Liu studied the queuing theory for traffic signal control and aimed to minimize total travel delay, using obtained performance values by queuing models [14]. Jiang et al. analyzed the tail lengths by modeling flow values with M/G/1 [15]. Shirmohammadi and Hadadi developed and applied the fuzzy controller to reduce the total delay and average queue length in urban intersections [16]. Giovanni et al. proposed a fuzzy-logic-based solution to manage traffic lights dynamically through taking into account the time of the day and the number of pedestrians about to cross the road [17]. Murat developed the model to estimate the average delay of vehicles by using artificial neural networks in isolated intersections [18]. Xu and Liu studied the differential evolution bacteria foraging algorithm to improve the traffic capacity of intersections in urban areas [19]. Babicheva described in another study the methods of queuing theory to optimize traffic signal phases on signal-controlled intersections [20]. Anokye et al. used the M/M/1 queuing model to get the performance of vehicular traffic on roads in city centers [21]. Prasad and Usha made a comparison of queuing models M/M/1 and M/D/1 for vehicular traffic at districts in urban areas of cities [22]. In another queuing analysis of traffic flow, Lee and Liu modeled traffic queuing with the same models, M/M/1 and M/D/1, and they suggested their models by increasing the number of lanes or adjusting green light time for each direction in busy hours during the day. This study stands out as taking into account the non-ergodic condition, unlike the previous queuing analyses [23]. There are also some studies that aim to improve the traffic light controller to manage phases, using connected sensors. Ghazal et al. designed a portable controller device that evaluates traffic density with IR sensors. They aimed to solve the problem of emergency vehicles in traffic jams [24]. Gundogan et al. developed the ATAK system as a traffic management tool at signalized intersections and they used genetic and fuzzy-logic algorithms to manage and control the isolated or coordinated intersections [25]. This system uses the number of vehicles from the input sensors placed at 50 and 200 m intervals at the intersection entrances and the output sensors placed at 10 and 20 m intervals at the exits. ATAK is currently and actively used in many of the central intersections of the city of Istanbul. In another traffic control system developed by ISSD Inc, unlike the ATAK system, traffic signal control is performed by taking vehicle images from camera systems [26]. The calculation method used in the system starts with drawing the graphs of the time-dependent changes of the vehicles passing through the intersection arms. Based on the graph curves created, the results of the 2nd and 3rd order polynomials obtained by curve fitting are compared with the initial values, then the appropriate green times are calculated.

One of the most important purposes of this research is to provide a solution to the traffic problem, which has become a serious issue in our cities where the urbanization rate is on the rise. With the actual observation and system data used, there have been attempts to show that an alternative method can be applied by the industry practitioners. With the data obtained from field sensors, the aim is to operate the traffic light durations and the phase transitions of connecting branches dynamically. The difference between the working principle to be applied dynamically and fixed systems was revealed and applied in the field. There are many points among the factors of this problem, such as junction locations, applied methods, used technologies, and sensors. These elements are explained and detailed in the following sections of the study. The key steps of this study are given in Figure 1 as flowchart.

The models suggested in this study were compared with fixed and traffic-sensitive operated intersections. Based on the performance values obtained by these proposed analytical methods, it was revealed that some intersections had a high intensity and the intersection efficiency needed improvement. Accordingly, it was observed that some traffic

flow directions were intense at certain periods of the day, and with an improvement in this regard, savings can be achieved in matters such as time, fuel, and energy. A dynamic phase sequence model was designed according to the results acquired, which allowed managing the intersection phase sequences dynamically.

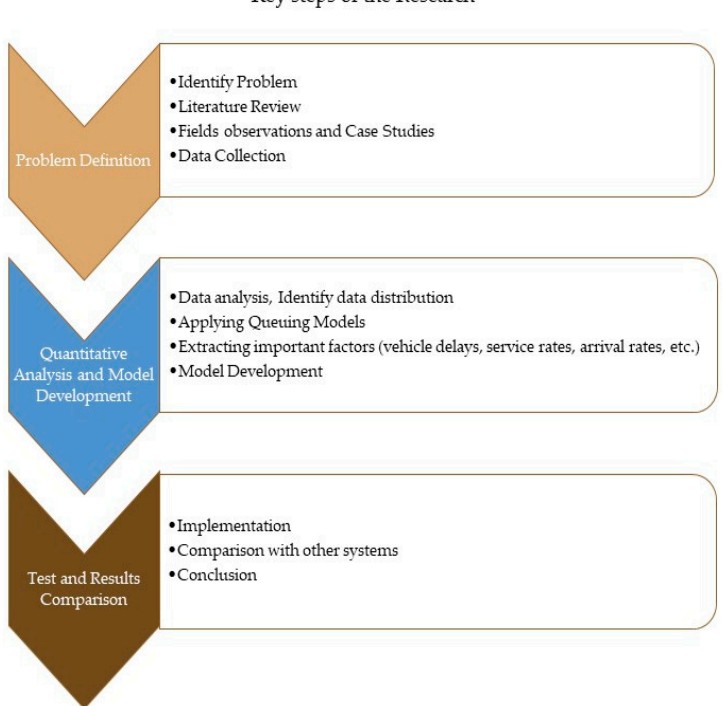

**Figure 1.** Key steps of the research methodology.

This model seeks to prevent loss by considering the density data of the intersection branches by assigning priority to these branches in the change of phases. The model was tested in three intersections according to old and new operating principles. The intensity and waiting times were lower in comparison to static and dynamic models, especially at peak hours of the day (refer to Section 3). Likewise, the calculation method of effective green time is suggested in Section 2.6. According to this method, it is aimed to produce an output based on the performance values obtained and to proportionally distribute the signal times accordingly. The positive results are shared in Section 3.

### 1.2. Intelligent Transportation System

Intelligent Transportation Systems (ITS), which emerged as a result of the adaptation of information and communication technologies to the transportation sector, follow a parallel development with these applications in the world. ITS integrate telecommunications, electronics, control theory, and computer technologies with the transportation sector [27]. In addition, the developing sensor and communication technology causes improvements in the infrastructure of the roads and brings out the concept of smart roads [28]. Accordingly, ITS technologies continue to spread rapidly.

Today, with the developing technology and fast mobility, many innovations are coming to the agenda in the transportation sector. Thanks to Intelligent Transportation Systems, ideal traffic conditions are created by ensuring coordination between different transportation types, and the efficiency and speed of the services related to passenger and vehicle movements are increased. Nowadays, smart cities and smart transportation are considered an inseparable whole [29]. It can be said that the management method of cities has changed with smart city technologies [30].

ITS provide significant benefits in the key points of the city with the data that they collect in areas with heavy traffic at hourly intervals. With these technologies, a well-organized traffic flow is achieved, and accidents and traffic jams can be prevented significantly. The use and correct positioning of these systems are very important, and, in this connection, central traffic management systems are one of the most important components of Intelligent Transportation Technologies in smart cities [27,30,31].

### 1.3. Traffic Management System

Smart cities use advanced traffic management systems that help to monitor, control, optimize and operate traffic in urban areas very effectively. The easiest way to manage and control traffic flows depends on sensing and monitoring road conditions. All these systems provide live data streaming to a traffic management center that, in turn, allows data transfer authorities and consumers to receive real-time updates on the city's transportation conditions and availability [31].

Traffic control centers or management systems serve as control rooms for smart city transport. They enable data, such as traffic status, congestion, speed, accident, flow, and density, to be intervened. This can regulate the entire traffic system, such as responding to current demand, reducing traffic congestion, and increasing safety. In summary, these systems are controlling systems that enable cities to adjust the flow of traffic as required. A variety of commonly used technologies can be connected to the traffic management center [31,32]. These can be control and monitoring units, such as road signs, classifiers, vehicle counters, traffic lights, camera sensors, etc. Figure 2 below shows an intersection plan that illustrates the sensors and vehicles of an intersection analyzed within the scope of this study.

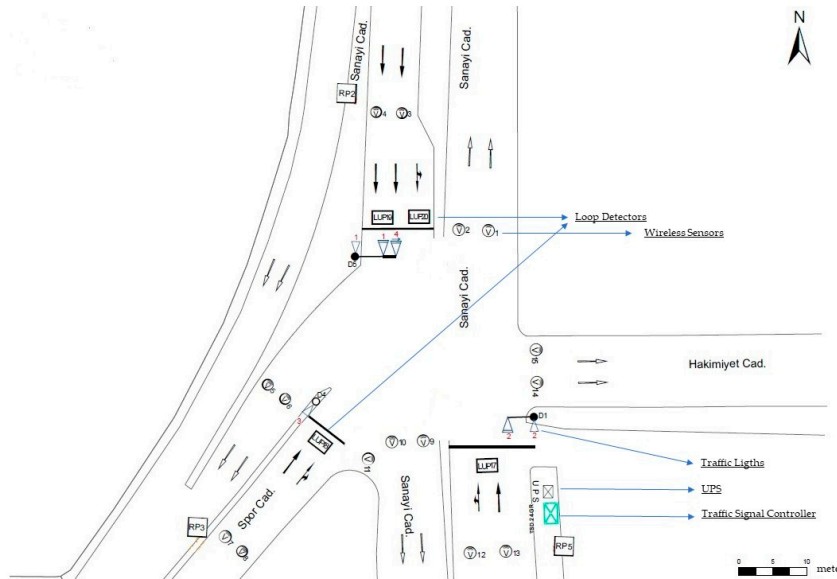

**Figure 2.** Intersection plan with monitoring units Istanbul–Kartal Intersection No 4.1 [33].

Many parameters are collected from the roads with loop detectors or camera sensors, such as the number of vehicles, weather conditions, density data, and momentary movements, leading to the dynamic management of traffic. With these developments, managers are particularly interested in accessing real-time data and using these data in decision support systems.

## 2. Materials and Methods

### 2.1. Signalized Intersections

Signalized intersections are located at the points of more than one connection road in the city center or on the highways. They are used as traffic regulators in city plans.

Since intersections are points of traffic flow, intersections that are incorrectly positioned may cause an increase in traffic intensity. Due to this sensitivity, the area should be well designed and well planned in order not to increase the traffic density and to prevent traffic accidents [34].

Traffic lights at intersections are operated by control devices. They enable road users to enhance their travels by assigning the right to take each approach and action. Traffic lights at signalized intersections may have more than one operating principle depending on the road condition, vehicle density, or connecting arms [34,35].

### 2.1.1. Signaling Plans

Signal operating modes can work in different time schedules and ways, such as pre-timed fixed time, semi-fixed, fully sensitive, hybrid, or adaptive [36]. Table 1 explains the signal modes with their definitions.

**Table 1.** Definitions of the signal plans.

| Signal Mode | Definition |
| --- | --- |
| Pre-timed fixed | Operates with fixed cycle times and green times. |
| Semi-fixed | Operates with fixed at certain times that do not vary much daily. |
| Sensitive | Operates by considering several parameters such as the number of vehicles, vehicle types, peak periods, etc. |
| Adaptive | Operates as a fully sensitive mode that controls also the phase priorities in addition to signal management. |
| Hybrid | It can work in different modes depending on the situation, such as peak hours, weather conditions, vacation days, etc. |

Pre-timed signaling systems operate with fixed cycle times and green times and apply the transitions between circuits in a fixed sequence. Signals that operate in this way can contain multiple timing plans with different cycle, split, and offset values for different periods of the day.

Semi-fixed or traffic-sensitive signal plans can be operated in a semi-fixed time or fully variable mode. The application in an intersection may vary depending on the capability of the sensors and the traffic planners' preference. Semi-fixed-term signal plans can work fixed at certain times that do not vary much daily and are sensitive to traffic at other times of the day. They are widely used in intersection management systems, as they are generally the preferred plan. Fully sensitive traffic signal plans, on the other hand, have a principle that can make instant planning by taking into account different parameters, such as the number of vehicles taken from the connection roads equipped with detectors, intensity, vehicle types, and peak periods. They can be designed to be able to plan according to the end of each period or a certain threshold value. Systems operating in this way try to evacuate the dense arm by changing the green, red, and protection times within the fixed circuit time [37]. Figure 3 shows the change plan showing the signal transitions [38].

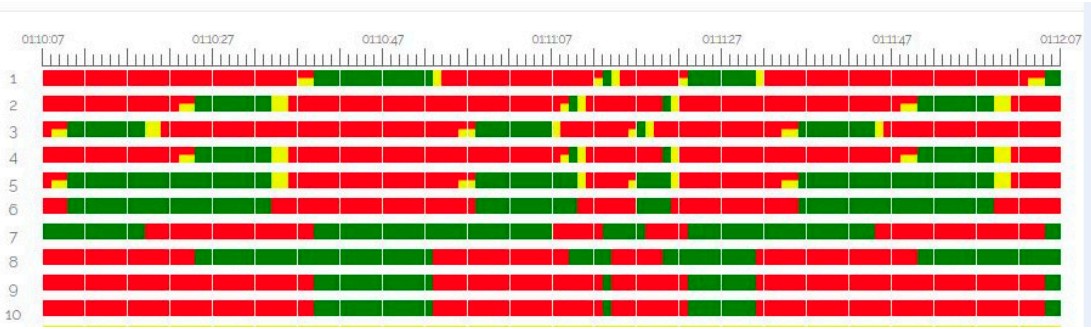

**Figure 3.** Transitions of signal plans [38].

Another signal plan is adaptive signal control. This plan is evaluated in the group of fully sensitive signal plans. Adaptive control is considered in a separate group due to one difference. In this plan, it is possible to give priority to the connected branches of intersections according to their intensity in addition to the signal time control. Right of way is granted by taking into account values, such as queue lengths, vehicle density, and vehicle types, on the intersection approach arms. Adaptive control can distribute green times evenly across all traffic flows, increase travel time reliability and extend the effectiveness of a traffic signal timing plan. For the system to work properly in this planning group, the data must be taken in real time or at certain intervals, and the intersection geometry must comply with this principle [39].

### 2.2. Vehicle Detection and Sensors at Intersections

Sensors are extremely important components for smart transportation systems. Their use enables the development of a wide variety of applications for traffic safety, traffic control, and driver assistance. The use of analytical methods in traffic flow problems was initiated with the use of sensors integrated with transportation systems. This integration has also paved the way for a wide variety of next-generation smart applications that aim to improve the security and traffic control of existing and future transportation systems.

It is extremely important to support the system management with the data flowing from the field to perform the system control in signalized intersections. By placing detectors or sensors inside the intersections or connecting roads, it becomes possible to control and plan the signal according to the number of vehicles and density ratios, or to pedestrian or bicycle activity. The traffic signal controller can then use this information to perform functions, such as allocating the amount of green time, selecting timing plans, and determining the sequence of transitions between phases.

Different sensors can be used in signalized intersections depending on factors such as intersection geometry, location, and capacity. Sensors are classified into two main groups according to their positioning: on-road/under asphalt (intrusive) or roadside at ground level (non-intrusive) [40]. Sensors such as cameras, radar, laser, infrared or ultrasonic are used roadside, while other types of sensors are used on the road/under asphalt, such as pneumatic road tubes, magnetic loops (loop detectors), and piezoelectricity. Intrusive sensors are placed under the asphalt or at road level [41]. They have high sensitivity and can be applied in a wide area, but at the same time, the installation effort, maintenance, and repair costs are quite high. Additionally, they are more affected by weather conditions (i.e., snow, rain, fog). Figure 4 shows the explained different types of these sensors.

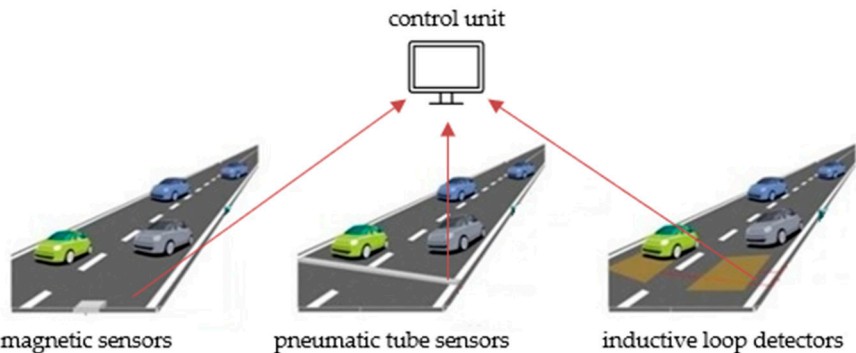

**Figure 4.** Intrusive sensors placed on or under asphalt [40].

Non-intrusive sensors are installed in different places as shown in Figure 5. These can see the road remotely and can detect parameters such as vehicle passage, speed, type, and lane capacity. These systems are more costly and more affected by environmental conditions compared to sensors located on asphalt. Remote, on-road sensors are mainly used to develop applications that provide information, such as queue detection at a traffic light, traffic conditions, road weather conditions, highway capacity, and current density.

Some sensors are mounted on a pole and used to monitor a specific coverage area, while others are mounted directly at bridge crossings above the road area to be monitored [40].

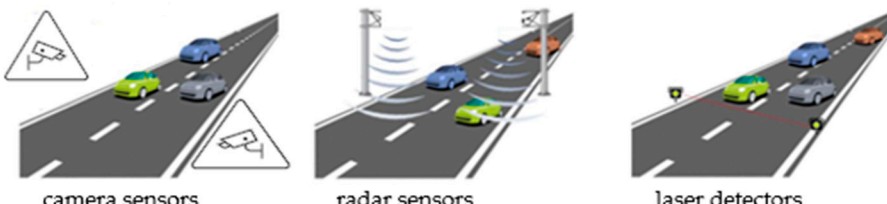

**Figure 5.** Non-intrusive sensors placed on the roadside at ground level or with a view of the road from above [40].

Acquiring the correct data flow is extremely critical in evaluating traffic conditions. This requires the use of the correct sensors in the correct locations and conditions. Positioning the sensors is done at the beginning with the intersection plans. Figure 6 shows the intersection plan made before the installation of the sensors. It has been noted that cameras and loop detectors are predominantly used at the intersections examined within the scope of this study. In this paper, there are indications that both systems have advantages and disadvantages in comparison. Moreover, different evaluation alternatives are formed according to the used technology. The resulting differences are mentioned in the discussion section of the study.

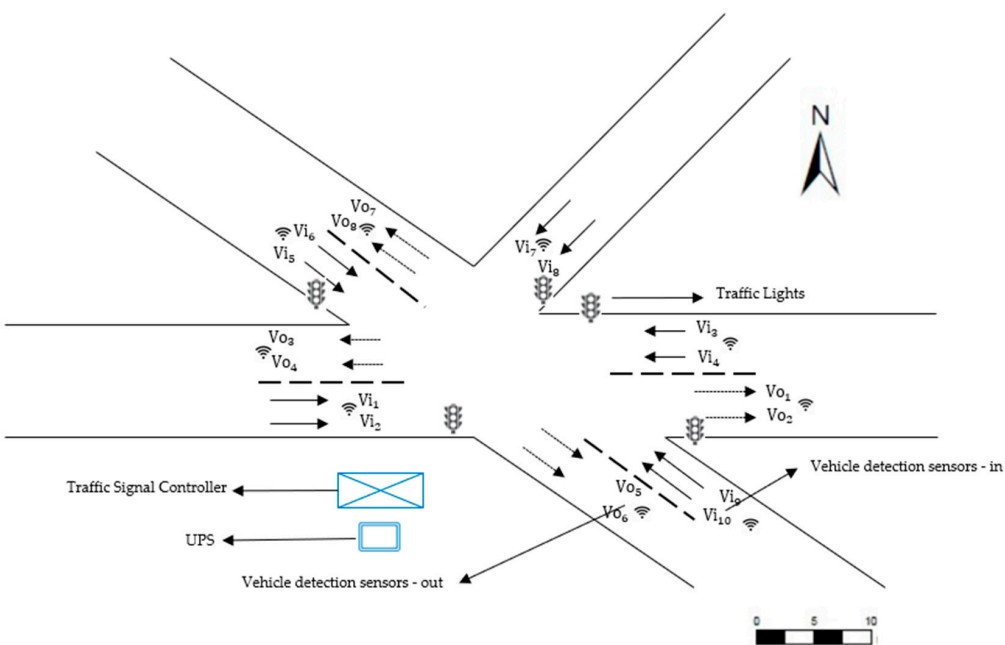

**Figure 6.** Sensors for vehicle counting placement plan on the intersection, Kocaeli Kartonsan [42].

### 2.3. Performance Measures of Signalized Intersection

Signalized intersections are important points or nodes in highways and urban transport systems. Evaluating a signalized intersection or suggesting a specific method to improve its operation is a challenging task. The level of service of the intersections can be evaluated with the help of simulation tools by using measurements such as capacity analysis or service time. As mentioned before, the most common criteria used as performance evaluation of signalized intersections are average delay per vehicle, average tail length, and the number of stops. The delay parameter in particular provides the most important value that defines both the performance of the intersection and the loss experienced by the vehicles. Latency is a measure of the excess time spent crossing the intersection that directly

correlates to the driver's experience [43]. The value of queuing at any time is extremely important in determining when to evacuate the buildup that may occur in a particular arm of an intersection. Another important measure used in determining intersection service quality is the number of stops as a valuable input parameter for environmental or air quality assessment models.

Datasets are needed to evaluate the intersection performance, using appropriate simulation models. Thanks to the developing infrastructure and sensor equipment, it is now possible to collect data from a certain intersection and its surroundings. The datasets collected may differ according to the capabilities of the sensors of the system. However, the collected data elements and the outputs of these data are listed as follows [44]:

- Number of vehicles in the intersection and approach arms;
- Vehicle types (automobile, truck, van, etc.);
- Number of pedestrians and bicycles;
- Physical road characteristics (number of lanes, degrees of approach, etc.);
- Green time of vehicles in the intersection;
- Average vehicle delays;
- Field observations in peak time zones;
- The arrival rate of approaching vehicles;
- Waiting times on the arms;
- The intensities of different intersections in nearby areas.

### 2.4. Queuing Models

In this study, queuing theory models are used to obtain performance values of intersections. The queuing theory allows mathematical analysis of queuing or waiting for lines that occur in any event. In particular, it is very convenient to calculate the queue lengths of the connected roads with the data provided by the intrusive sensors on the intersection connecting arms. Queuing models are expressed with notations, given in Table 2 below, such as M/M/1, M/M/c, M/G/1, or G/G/1, depending on some factors, such as the distribution of the data examined, service points, and source of incoming data [45].

**Table 2.** Definitions of the terms used by queue models.

| Notation | Description |
|---|---|
| M | Markov property is a stochastic process and satisfies that arrival and departures have Poisson distribution. In queue models, the first value denoted by M indicates arrivals, while the second value indicates departures [45]. |
| G | This notation means that the arrival and departures have arbitrary distribution and are denoted by G (general). |
| c | Service channel or points. |

In the articles prepared before this study, queues at intersection points were analyzed with different queuing models, and performance values were derived accordingly [46]. Since the subject related to queuing theory models and its application in signalized intersections in detail is mentioned in different studies by the same author group, this subject will not be detailed in this study. However, since the outputs of this study are generally obtained by the queuing theory, the parameters and variables used will be mentioned.

However, note that the calculations in Table 3 are given according to the M/M/1 model. If no assumptions are made, the queue model notations change according to the vehicle arrival and departure distribution characteristics [45,46]. Accordingly, some of the calculations given above are obtained with the formulations given below. In the M/G/1 model, the average number of vehicles in the queue is calculated by using Equation (1), known as the Pollazcek–Khintichine formula [47].

$$L_q = \frac{\lambda^2 \sigma^2 + \rho^2}{2(1 - \rho)} \tag{1}$$

**Table 3.** Description of relevant parameters of queuing models.

| Variable | Definition/Calculation | Description |
|:---:|:---:|:---:|
| $\lambda$ | Arrival rate | The average number of vehicles arriving per unit time |
| $\mu$ | Service rate | The average number of vehicles served per unit of time |
| $\rho$ | $\lambda/\mu$ | Intensity (flow ratio) |
| $L_s$ | $L_s = \frac{\lambda}{\mu-\lambda}$ | The average number of vehicles in the system |
| $L_q$ | $L_q = \frac{\lambda^2}{\mu(\mu-\lambda)}$ | The average number of vehicles in the queue |
| $W_s$ | $W_s = \frac{\rho}{(1-\rho)\lambda}$ | Average time spent in the system |
| $W_q$ | $W_q = \frac{\rho}{(\mu-\lambda)}$ | Average time spent in the queue |

However, for the G/G/1 model, the following approximation is proposed by Marchall, where $\sigma_a^2$ is variance of interarrival times and $\sigma_s^2$ is the variance of service times [48].

$$C_a^2 = \frac{\sigma_a^2}{\left(1/\lambda\right)^2} \tag{2}$$

$$C_s^2 = \frac{\sigma_s^2}{\left(1/\mu\right)^2} \tag{3}$$

$$L_q \approx \frac{\rho^2\left(1 + C_s^2\right)\left(C_a^2 + \rho^2 C_s^2\right)}{2(1-\rho)(1+\rho^2 C_s^2)} \tag{4}$$

*2.5. Dynamic Phase Sequences Model for Signalized Intersection*

2.5.1. Scheduling Problem

One of the most important issues in intersection control is the efficient use of signal planning through the traffic control device. If the intersection is designed to operate in a dynamic structure, the intersection working order can be improved by using some timing methods on the signal controller. According to the intensity or occupancy rates to be achieved from each connected arm, transactions can be taken without allowing a queue to form with a certain ranking or scheduling algorithm. Algorithms used to run more than one transaction in a certain order are generally classified as preemptive and non-preemptive. The main difference here is that a system does not experience any intervention until the process is terminated or the process is interrupted by a priority vehicle (emergency service vehicle, VIP platoon) while the process is in progress. There are some prominent algorithms used in operating systems or related areas regarding process scheduling. These are methods that can be adapted according to the appropriate scenario with different operating principles [49,50].

We can list the main process scheduling algorithms used as follows:

- Shortest Job First (SJF);
- Round-Robin (RR);
- First Come First Served (FCFS);
- Priority Scheduling Highest Priority First (HPF);
- The Shortest Remaining Time (SRT).

For these methods to be used in signalized intersections, which are the most important component in regulating the traffic flow, the working principle of the scheduling algorithms should be analyzed well. All methods may have advantages or disadvantages depending on the case study. In this article, not all of the scheduling algorithms mentioned above are explained in detail.

Furthermore, the main subject of this study is not to compare these algorithms or to elaborate their prominent side. However, it is highlighted why the appropriate method was

chosen and how it was adapted to the problem. As mentioned above, not every method can be adapted to the examined problem. For example, the SJF method focuses primarily on finishing the shortest job. It aims to complete the shortest or simplest job at hand and finish the next shortest job. Thus, the work will soon be continued by the assigned priority order. If this method is applied to an intersection problem, the following problem may arise. If the ratio of the three arms is always shorter or less than the value of the fourth arm, there is a possibility that the busiest arm will not be processed at all. Therefore, this model can complete short jobs quickly, but it can also cause busy arms to be even more intense at the same time.

One of the most important outputs of this study is the dynamic phase sequence model, and a priority-based scheduling algorithm is used in this proposed model. The basic logic in the priority-based scheduling algorithm is to serve the units during the operation according to their priority value. Especially in real-time systems, some processes must be completed within a specified period. The aim is to complete the processes without exceeding the service capacity rather than the time they are planned, started, or finished. The target in this planning is to complete the operation of the processes without exceeding the maximum service time.

Based on the similarities of this method, it is aimed to determine the working order of the branches and to work dynamically by creating priority values according to the critical current values calculated in signalized intersection connections. Thus, it is proposed to control the phases without the occupancy rate in the tail arms exceeding the critical value.

### 2.5.2. Proposed Model for Dynamic Phase Sequences

In this study, a part of the Priority Scheduling method, which is briefly described under the title of scheduling algorithms, is used to propose the problem related to the phase order in intersections as mentioned above. Here, this method was used only to prioritize the phases. The main starting point of this study is determining the intensity of the intersection arms according to the number of vehicles obtained by the sensors with a queue analysis. Therefore, since the detailed description of scheduling algorithms may be the subject of a different study, it is not detailed here.

It is important to prioritize busy arms in terms of phase management in intersection control. When the graphs obtained in the results section are examined, it is seen that the intersections cannot manage the incoming flows regularly and the vehicles wait for each other disproportionately. The proposed flow diagram in Figure 6 aims to order the phases regarding intensity. Before the flow of the diagram, the parameters used and their definitions are given in Table 4.

Based on these results, a dynamic phase sequence model is offered. The flow diagram of the method for the dynamic phase sequences method is given in Figure 7.

The queue performance was explored by obtaining values such as intensity, time spent by vehicles, number of vehicles, and queue lengths at intersections. Here, it is considered that the inter-arrival times of incoming vehicles are exponentially distributed. When the queuing theory is applied, it is generally investigated that the intensity data are $\rho < 1$. In cases where the service rates ($\mu$) are lower than the arrival rates ($\lambda$) of the vehicles, it is inevitable for the relevant waiting line to form a queue. For this reason, in such cases, it is possible to prioritize and unload the relevant queue line when it reaches a certain threshold value.

### 2.6. Dynamic Signal Timing Method for Intersections

The signal operating modes used in signalized intersections are the most important parameters affecting the performance of the intersections. As mentioned in Section 2.1.1, signal models can have fixed, semi-constant, or fully actuated (dynamic) modes. These modes are extremely important in the efficiency of intersections and in managing traffic.

**Table 4.** Description of relevant variables.

| Variable | Unit | Description |
|---|---|---|
| $k$ | - | Number of incoming vehicles |
| $t_i$, $t_{i+1}$ | Entry times of the vehicles to the intersection | Arrival times of the vehicles |
| $l_i$ | - | Phases of intersections |
| $N_t$ | s | Difference between arrival times |
| $S_t$ | s | Difference between service times |
| $Q_{kl}$ | - | Number of vehicles arriving per unit time<br>This parameter is also shown with $\lambda$ on queuing models. |
| $Q_{sl}$ | - | Number of vehicles serviced per unit time<br>This parameter is also shown with $\mu$ on queueing models. |
| $\varphi_l$ | - | Traffic flow rate (flow intensity)<br>This parameter is also shown with $\rho$ on queueing models. |
| $\gamma$ | - | Saturation flow rate (maximum flow rate highest traffic intensity) |
| $\varphi_d$ | - | The highest current density is seen during the day. |

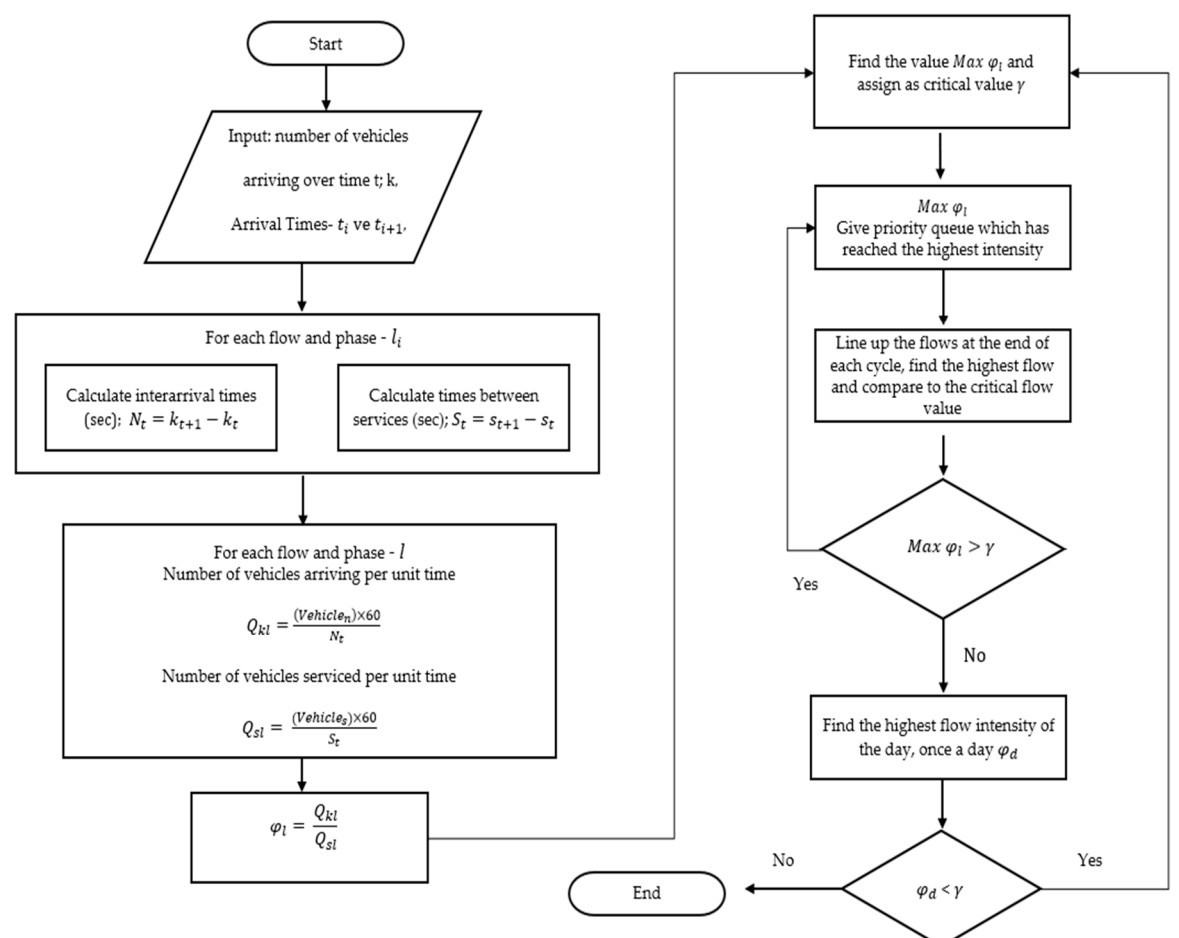

**Figure 7.** Flow diagram of dynamic phase sequences model for a signalized intersection.

In this study, the intersections chosen for the model study are the intersections that can provide data, thanks to the sensors that they have. Therefore, values such as time between phases or currents, circuit duration, saturated flow, red, green, and protection times are quite sufficient to propose a new model. Green times are critical values that affect queuing in the junction arms or the discharge of the queue lengths. As a result of the analysis and the gained outputs, a calculation method for effective green periods is proposed in this study. It is aimed to calculate the effective green times and to give more time to the

busy branches by using the critical current values calculated over the number of vehicles taken from the intersection connecting arms with the proposed signal durations calculation method. Therefore, it is aimed to use proportionally more regular green time in the periods when the connected arms are intense and to prevent unbalanced waiting losses.

In the effective green time calculation method, the critical current value calculated in the dynamic phase sequence model suggested in Section 2.5.2 and the time created by the intersection at the end of each cycle were taken as the main input.

The number of vehicles arriving per unit time:

$$Q_{kl} = \frac{Vehicle_n \times 60}{N_t} \tag{5}$$

The number of vehicles passing through the traffic light point per unit time:

$$Q_{sl} = \frac{Vehicle_s \times 60}{S_t} \tag{6}$$

By finding the ratio of these values to the saturated critical flow, it is aimed to distribute the green times proportionally to all four arms. The flow of the method is given in Table 5. As can be seen from the table below, the new critical value is determined by comparing the flow intensities with the highest saturated flow value at the end of the day.

**Table 5.** Process steps for calculation of effective green time.

| |
| --- |
| **Input:** $Q_{kl}$, **Arrival Rate;** $Q_{sl}$, **Service Rate;** $\rho$, **Flow Intensity;** $C$, **Cycle Time;** $\varphi_i$, **the Maximum Value of Intensity;** $\phi$, **Assigned as a Critical Value of Highest Intensity;** $G_i^k$, **the Green Time for Each Phase;** $T_l$, **Loss Time;** $Y_t$, **Yellow Time** |
| Output: $G_e$, the effective green time for each phase |
| Steps. <br> 1.　Find the number of vehicles arriving per unit time for each approach branch of the intersection, Arrival Rate $Q_{kl}$, and find the number of vehicles served per unit time for each approach branch of the intersection, Service Rate $Q_{sl}$. <br> 2.　Calculate flow intensity $p = Q_{kl}/Q_{sl}$. <br> 3.　Calculate the intensity for each phase $\varphi_i$. <br> 4.　Assign the maximum value of $\varphi_i$ obtained at the end of each cycle as critical value $\phi$. <br> 5.　Find cycle time, $C$. <br> 6.　Calculate the green time for each phase using the circuit duration, intensity value, and critical value $(k)$. <br> $G_i^k = \frac{C}{\sum_{i=1}^{k} \frac{p_i}{\varphi_i}}$, $G_i = G_i^k \times p_i$ <br> 7.　Calculate effective green time. <br> $T_l = Loss\ Time$, $Y_t = Yellow\ Time$ <br> $G_e = G_i + T_l - Y_t$ <br> 8.　At the end of each cycle, find the critical rate again and compare it with the previous critical value. If $(\varphi_{ij} > \varphi_i)$ go to step 4, else go to step 6. <br> 9.　Find effective green time for each phase. <br> 10.　End. |

### 2.7. Data Collection and Study Area

The data acquisition stage, which is one of the most important inputs of the research, was carried out in three different provinces at intersections with different infrastructures. Within the scope of the study, field studies were first conducted on the intersections selected from Istanbul and Kocaeli provinces in Turkey. Later, this scope was extended with the intersections selected from Konya.

The working principle of the intersections in Konya province differs from the others due to the sensors used. While loop detectors and wireless sensors are mainly used in Istanbul and Kocaeli, the vehicle detection systems in Konya work with camera sensors. The main difference here is that sensors positioned on or below asphalt are positioned

at certain distances on the junction link arms, while the cameras are positioned close to the intersection center to detect vehicles passing through the intersection endpoint. This fundamental difference in sensor systems require performance values to be calculated with different analysis methods. For example, while the tail lengths on the junction approach arms can be calculated more accurately with the loop detectors, this could be calculated with a slightly more complex method with the camera sensors. However, this is not a measure of the need to use a camera as a vehicle sensor at intersections. As mentioned in Section 2.2, since both systems may have pros and cons concerning each in general terms, the preference of the system to be used may vary according to the parameters prioritized by the practitioners and city managers. Figure 8 shows the image of the intersection equipped with a camera system and examined in this study.

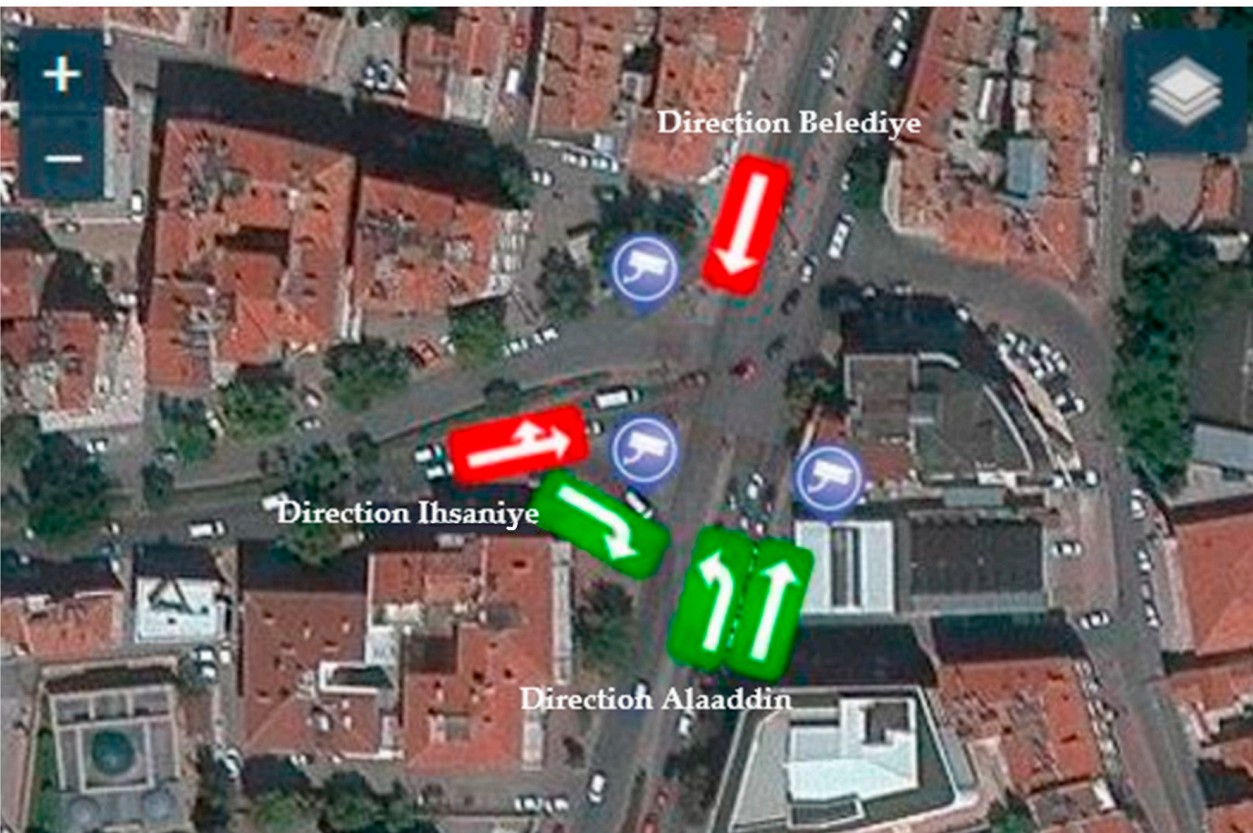

**Figure 8.** Junction view equipped with camera sensor (Konya/Turkey) [33].

While the selected intersections in Istanbul and Kocaeli are isolated, that is, the intersections do not have any connection or influence with any intersection near them, the intersections in Konya covering both situations were examined. The main purpose here was to see the relationship between two close intersections and to examine the interaction between them. However, the interaction between close intersections is planned to be conducted as a different research subject; hence, it is not within the scope of this study. In this study, only isolated intersections were evaluated.

It should be noted that the investigations carried out in the selected provinces were carried out on sunny days and in the same period. The data of January and March were studied as a period. In the studies carried out in Istanbul and Konya provinces, 3 junctions with 3 and 4 connected arms were examined. These intersections are also considered to have both a fixed and dynamic signal pattern. The collected data were analyzed in different time intervals in 1 day and 10 day periods. Table 6 describes the analysis periods and hour intervals.

**Table 6.** Analysis periods.

| Hours | In Two Period (Period 1: 1 Day; Period 2: 10 Days) |
| --- | --- |
| 07:00–18:00 | 3 h interval: 07:00–10:00, 11:00–14:00, 15:00–18:00 |
| 07:00–18:00 | 15 min interval: 07:00–07:15, 07:15–07:30 . . . 17:45–18:00 |
| 07:00–18:00 | 1 h interval: 07:00–08:00, 08:00–09:00 . . . .17:00–18:00 |

The most important factor in creating signal plans for intersections is the management of phase transitions in a certain order. This is an important event in providing traffic flow. The diagram in Figure 9 shows the order of phases.

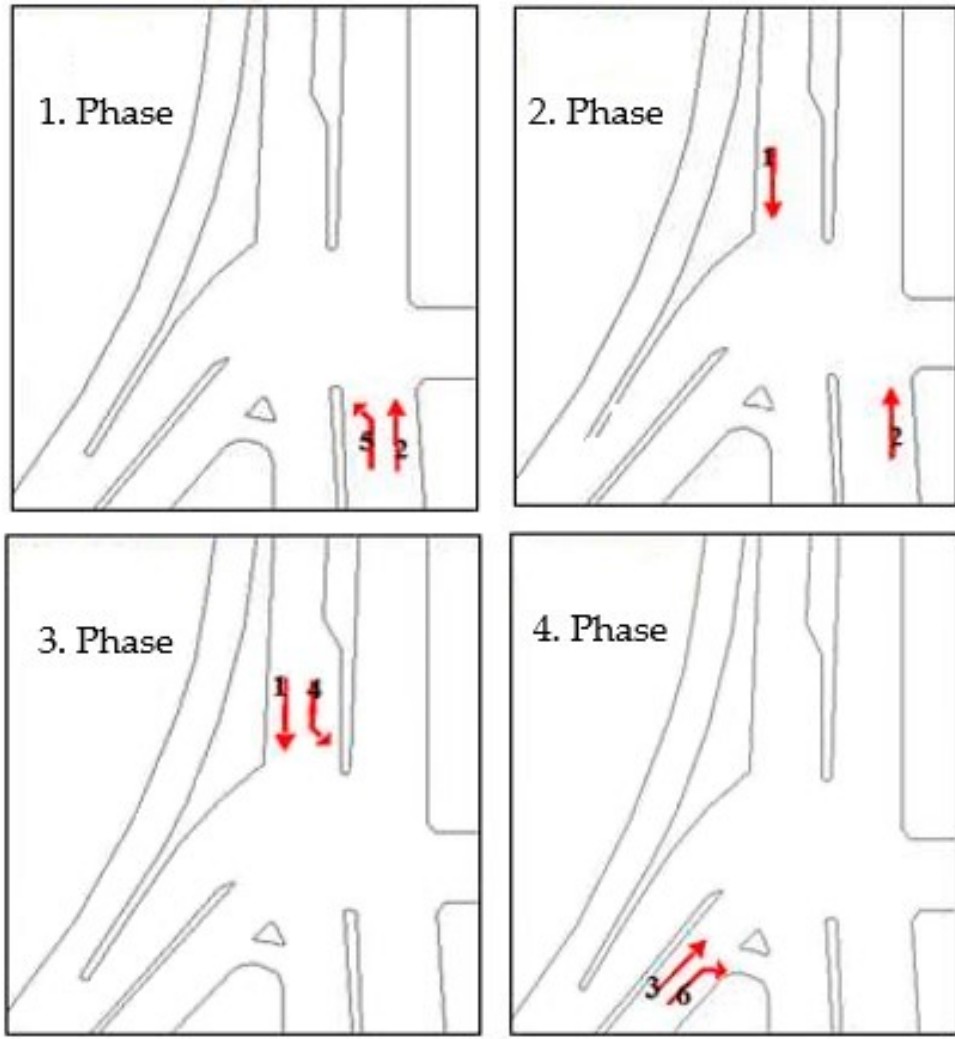

**Figure 9.** Phase transitions order of signal plans.

The information on which vehicle turned to which side from these data is known from the order of the phases. From this dataset, the information about the number of vehicles driving in the relevant intersection in which the phase is obtained from the difference of the values was given by the input–output sensors. In the same way, criteria such as the flow rate of incoming vehicles or service rates compared to the arrival rates were calculated over the input–output sensors. The dataset given as an example below was obtained from wireless sensors and loop detectors placed under asphalt. These types of sensors here give us the number of vehicles passing over. Sensors shown in red give the number of vehicles approaching the intersection, while green data give the number of vehicles exiting through the intersection.

Before mentioning the values of the sensors below, it is necessary to explain the phase, cycle, and current of the intersections. The meaning of the phase is explained above and Figure 9 shows the phases with their direction of rotation. Table 7 explains the terms that we will use very often in the study.

**Table 7.** Definitions of the terms used in intersections.

| Term | Description |
|------|-------------|
| Phase | Each phase in an intersection has a set of timings that control the transit times of pedestrians and vehicles using the intersection. One phase can control both the right turns of vehicles and pedestrian crossings [51]. |
| Flow | The concept of flow is traffic flow generated by vehicles approaching an intersection from any direction. |
| Cycle | A cycle is defined as the total time to complete a series of signaling for all movements in an intersection [51]. |
| Effective Green Time | The time spent by a particular traffic movement or movement sequence. |

In the analysis of the transition between phases, the changes in the data were taken as they are. The collected data were calculated by passing through some preliminary stages to be used in the modeling phase. For example, in Table 8, where a part of the dataset of the Kartal Sanayi Street intersection is shared, the values taken from the entrance sensors of the vehicles approaching the intersection are shown in red, and the values taken from the exit sensors in green. These values give us the number of vehicles passing through loop sensors placed under asphalt from V1 to V14.

**Table 8.** Example dataset collected from sensors of the intersections equipped with intrusive sensors.

| Phase | Time | Phase Time (sec) | V1 | V2 | V3 | V4 | V5 | V6 | V7 | V8 | V9 | V10 | V11 | V12 | V13 | V14 |
|-------|------|-----------------|----|----|----|----|----|----|----|----|----|-----|-----|-----|-----|-----|
| 1 | 00:00:02 | 130 | 3 | 4 | 1 | 0 | 0 | 0 | 0 | 0 | 0 | 0 | 0 | 0 | 0 | 0 |
| 2 | 00:00:15 | 160 | 0 | 0 | 0 | 1 | 0 | 0 | 4 | 1 | 4 | 2 | 0 | 1 | 1 | 0 |
| 3 | 00:00:31 | 131 | 0 | 0 | 0 | 1 | 0 | 0 | 0 | 0 | 0 | 1 | 0 | 6 | 5 | 0 |
| 4 | 00:00:44 | 139 | 3 | 3 | 1 | 0 | 0 | 0 | 2 | 0 | 0 | 0 | 0 | 5 | 5 | 0 |
| 4 | 00:01:01 | 50 | 1 | 2 | 0 | 0 | 0 | 0 | 0 | 0 | 0 | 0 | 0 | 2 | 1 | 0 |
| 1 | 00:01:06 | 130 | 1 | 4 | 1 | 0 | 0 | 0 | 2 | 0 | 0 | 0 | 0 | 1 | 2 | 0 |
| 2 | 00:01:19 | 220 | 11 | 9 | 1 | 1 | 0 | 0 | 3 | 2 | 2 | 2 | 0 | 0 | 0 | 0 |
| 3 | 00:01:41 | 131 | 3 | 0 | 0 | 1 | 0 | 0 | 3 | 1 | 0 | 0 | 0 | 1 | 0 | 1 |
| 4 | 00:01:54 | 79 | 1 | 0 | 1 | 0 | 0 | 0 | 2 | 2 | 0 | 0 | 0 | 1 | 0 | 0 |
| 4 | 00:02:05 | 50 | 1 | 0 | 0 | 0 | 0 | 0 | 2 | 0 | 0 | 0 | 0 | 4 | 2 | 0 |
| 1 | 00:02:10 | 130 | 4 | 5 | 0 | 0 | 0 | 0 | 2 | 2 | 0 | 0 | 0 | 4 | 4 | 0 |
| 2 | 00:02:23 | 190 | 8 | 4 | 1 | 1 | 0 | 0 | 2 | 2 | 0 | 1 | 0 | 7 | 3 | 2 |
| 3 | 00:02:42 | 131 | 1 | 0 | 1 | 1 | 0 | 0 | 2 | 1 | 0 | 2 | 0 | 1 | 0 | 1 |
| 4 | 00:02:55 | 79 | 1 | 0 | 0 | 0 | 0 | 0 | 1 | 1 | 1 | 1 | 0 | 0 | 0 | 0 |
| 4 | 00:03:06 | 50 | 2 | 1 | 1 | 0 | 0 | 0 | 0 | 0 | 0 | 0 | 0 | 0 | 0 | 0 |
| 1 | 00:03:11 | 130 | 4 | 4 | 0 | 0 | 0 | 1 | 0 | 2 | 0 | 0 | 0 | 1 | 1 | 0 |
| 2 | 00:03:24 | 191 | 7 | 3 | 1 | 2 | 0 | 0 | 6 | 2 | 0 | 1 | 0 | 6 | 3 | 0 |
| 3 | 00:03:43 | 130 | 1 | 0 | 1 | 2 | 0 | 0 | 0 | 1 | 0 | 2 | 0 | 4 | 3 | 0 |

Another dataset given in Table 8 shows a sample data group taken from an intersection equipped with a camera sensor. Values such as cycle periods, green times, protection times, and time differences between transitions, indicated in red in Table 9, were obtained. These data are important inputs for the created models.

**Table 9.** Data example of the phase coming from Adalhan Junction Alaaddin direction (sec).

| Date | Phase Diff. (sec) | Minute | Cycle Time (sec) | Green Time(sec) | Red and Yellow Time(sec) | Direction (Number of Vehicles) | Total Vehicle Count |
|---|---|---|---|---|---|---|---|
| 01/01/2020 06:59:13 | 419.22 | 1.23 | 74 | 16 | 58 | Arrivals from Alaaddin (8) | 8 |
| 01/01/2020 07:00:27 | 420.45 | 1.15 | 69 | 12 | 57 | Arrivals from Alaaddin (4) | 4 |
| 01/01/2020 07:01:36 | 421.60 | 1.23 | 74 | 12 | 62 | Arrivals from Alaaddin (2) | 2 |
| 01/01/2020 07:02:50 | 422.83 | 1.37 | 82 | 22 | 60 | Arrivals from Alaaddin (2) | 2 |
| 01/01/2020 07:04:12 | 424.20 | 1.38 | 83 | 15 | 68 | Arrivals from Alaaddin (5) | 5 |
| 01/01/2020 07:05:35 | 425.58 | 1.45 | 87 | 13 | 74 | Arrivals from Alaaddin (4) | 4 |
| 01/01/2020 07:07:02 | 427.03 | 1.28 | 77 | 12 | 65 | Arrivals from Alaaddin (0) | 0 |
| 01/01/2020 07:08:19 | 428.32 | 1.23 | 74 | 15 | 59 | Arrivals from Alaaddin (1) | 1 |
| 01/01/2020 07:09:33 | 429.55 | 1.22 | 73 | 12 | 61 | Arrivals from Alaaddin (2) | 2 |
| 01/01/2020 07:10:46 | 430.77 | 1.18 | 71 | 13 | 58 | Arrivals from Alaaddin (4) | 4 |
| 01/01/2020 07:11:57 | 431.95 | 1.18 | 71 | 12 | 59 | Arrivals from Alaaddin (2) | 2 |
| 01/01/2020 07:13:08 | 433.13 | 1.15 | 69 | 12 | 57 | Arrivals from Alaaddin (1) | 1 |
| 01/01/2020 07:14:17 | 434.28 | 1.20 | 72 | 14 | 58 | Arrivals from Alaaddin (2) | 2 |
| 01/01/2020 07:15:29 | 435.48 | 1.33 | 80 | 13 | 67 | Arrivals from Alaaddin (4) | 4 |
| 01/01/2020 07:16:49 | 436.82 | 1.15 | 69 | 13 | 56 | Arrivals from Alaaddin (0) | 0 |
| 01/01/2020 07:17:58 | 437.97 | 1.17 | 70 | 12 | 58 | Arrivals from Alaaddin (2) | 2 |
| 01/01/2020 07:19:08 | 439.13 | 1.42 | 85 | 14 | 71 | Arrivals from Alaaddin (3) | 3 |

## 3. Results and Discussion

Values such as vehicle delays, intensity, and queue lengths are critical parameters in evaluating the capacity and service levels of intersections. Among these parameters, the most frequently used vehicle delay value can be obtained in several ways depending on the infrastructure of the intersection or environmental factors. For example, in a four-leg junction, there is a possibility that vehicles coming from different points can cross in all four directions. The total elapsed time for a vehicle entering the intersection is the time spent from the moment a vehicle, coming from any current, enters the intersection arm until it leaves the intersection. In an intersection equipped with sensors, it is possible to obtain information about when the vehicle enters and leaves the intersection, how much time the vehicle spends at the intersection or how many vehicles are in the queue. As mentioned above, the queuing theory is used to make these calculations.

Before the intersection intensities and delays are calculated by the queuing theory, it is necessary to examine the distribution of the number of vehicles during the day. Since the intersections within the scope of the study were equipped with sensors, the number of vehicles was examined in detail first. Therefore, as can be seen in the graphic below, the change in the number of vehicles during the day was analyzed through graphs. Figure 10 below shows the number of vehicles coming from three different directions of the intersection, the sketch of which is given in Figure 8.

Similarly, these graphs were drawn for all the examined intersections to investigate the status of the intersections.

The important issue in queue models is to determine the distribution characteristics of arriving vehicles and vehicles passing through the service point. The notation and calculations of the queue model change according to these distribution characteristics [52]. Since the vehicle queuing models at the selected intersections were examined in detail in the previous study, the outputs obtained in this study were mentioned [46]. According to the formulas results are given in Table 10 below.

In the same analysis performed at the Kartal–Sahilyolu intersection in Istanbul, the performance of the phases, rather than the currents, was examined. Table 11 shows the results obtained within the scope of the evaluation made in 10 days. Examining the phases shows that the number of vehicles in the system and the mean time spent in some periods of the day are relatively high.

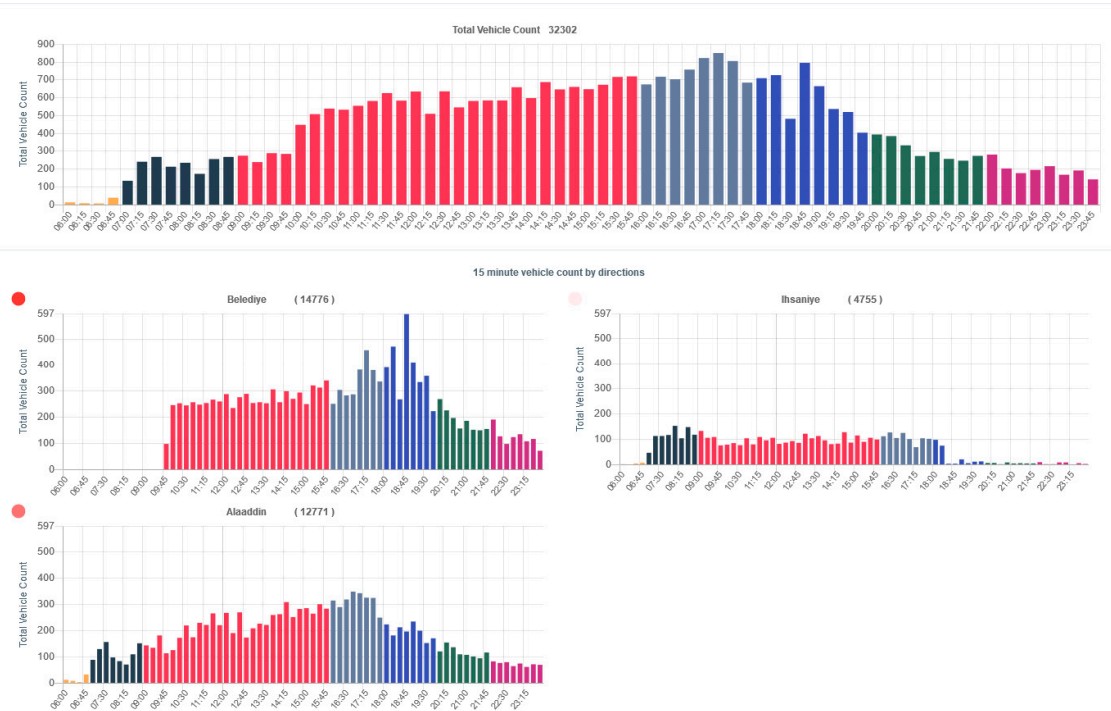

**Figure 10.** Number of vehicles comes from all directions to the intersection.

**Table 10.** The 1 day performance values at Kartal Sahilyolu Junction.

| Flow | Period (Hours) | Arrival Rate | Service Rate | Intensity (Flow Ratio) | Vehicle in the System | Mean Time Spent in the System (min) | Mean Time Spent in Queue (min) |
|------|----------------|--------------|--------------|------------------------|-----------------------|-------------------------------------|--------------------------------|
| 1. flow | 07–10 | 7.23 | 7.42 | 0.97 | 38.05 | 5.26 | 5.13 |
| 2. flow | 07–10 | 7.03 | 8.21 | 0.86 | 5.96 | 0.85 | 0.73 |
| 3. flow | 07–10 | 6.89 | 8.56 | 0.80 | 4.13 | 0.60 | 0.48 |
| 1. flow | 11–14 | 6.32 | 7.02 | 0.90 | 9.03 | 1.43 | 1.29 |
| 2. flow | 11–14 | 6.27 | 9.23 | 0.68 | 2.12 | 0.34 | 0.23 |
| 3. flow | 11–14 | 6.15 | 12.32 | 0.50 | 1.00 | 0.16 | 0.08 |
| 1. flow | 15–18 | 7.52 | 8.15 | 0.92 | 11.94 | 1.59 | 1.46 |
| 2. flow | 15–18 | 7.45 | 14.23 | 0.52 | 1.10 | 0.15 | 0.08 |
| 3. flow | 15–18 | 7.03 | 7.42 | 0.95 | 18.03 | 2.56 | 2.43 |

**Table 11.** The 10-day performance values of phases at Kartal–Sahilyolu intersection.

| Phase | Period (Hours) | Arrival Rate | Service Rate | Intensity (Flow Ratio) | Vehicle in the System | Mean Time Spent in the System (min) | Mean Time Spent in Queue (min) |
|-------|----------------|--------------|--------------|------------------------|-----------------------|-------------------------------------|--------------------------------|
| 1. Phase | 07–10 | 5.32 | 5.45 | 0.98 | 40.92 | 7.69 | 7.51 |
| 2. Phase | 07–10 | 2.29 | 2.65 | 0.86 | 6.36 | 2.78 | 2.40 |
| 3. Phase | 07–10 | 2.98 | 4.52 | 0.66 | 1.94 | 0.65 | 0.43 |
| 4. Phase | 07–10 | 3.52 | 6.3 | 0.56 | 1.27 | 0.36 | 0.20 |
| 1. Phase | 11–14 | 4.72 | 5.63 | 0.84 | 5.19 | 1.10 | 0.92 |
| 2. Phase | 11–14 | 0.75 | 1.2 | 0.63 | 1.67 | 2.22 | 1.39 |
| 3. Phase | 11–14 | 2.74 | 2.96 | 0.93 | 12.45 | 4.55 | 4.21 |
| 4. Phase | 11–14 | 4.23 | 5.52 | 0.77 | 3.28 | 0.78 | 0.59 |
| 1. Phase | 15–18 | 8.69 | 10.35 | 0.84 | 5.23 | 0.60 | 0.51 |
| 2. Phase | 15–18 | 1.17 | 2.1 | 0.56 | 1.26 | 1.08 | 0.60 |
| 3. Phase | 15–18 | 3.52 | 3.68 | 0.96 | 22.00 | 6.25 | 5.98 |
| 4. Phase | 15–18 | 4.56 | 6.85 | 0.67 | 1.99 | 0.44 | 0.29 |

We can calculate the results below for the 1st phase according to the formulas given in Table 3. The number of vehicles obtained from the entrance sensors was 957 in a 3 h period for the 1st phase, and the number of vehicles obtained from the output sensors in approximately 79 minutes of green time was 431.

For the number of vehicles arriving per unit time, the arrival rate is as follows:

$$\lambda = \frac{957}{180} = 5.32 \tag{7}$$

For the number of vehicles serviced per unit time, the service rate is as follows:

$$\mu = \frac{430}{79} = 5.45 \tag{8}$$

For the intensity, the flow ratio is as follows:

$$\rho = \frac{5.32}{5.45} = 0.98 \tag{9}$$

The number of the vehicles in the system is as follows:

$$L_s = \frac{\lambda}{\mu - \lambda} = 40.92 \tag{10}$$

The mean time spent in the system and queue is as follows:

$$W_s = \frac{\rho}{(1 - \rho)\lambda} = 7.69 \tag{11}$$

$$W_q = \frac{\rho}{(\mu - \lambda)} = 7.51 \tag{12}$$

In the following Figure 11, the variation of the average vehicle durations obtained from the phases during three different periods of the day can be seen. The figure below demonstrates the average vehicle delays experienced in four different phases during three different periods of the day. The fluctuation of each phase in separate time zones indicates that the operating principles of the phases need improvement. In a well-planned intersection, the time differences between vehicle delays should not be too long due to the phase duration. For example, the average time spent by vehicles in the 1st phase between 15:00–18:00 is 0.6 min, while in the 3rd phase it goes up to 6.25 min. Likewise, while the delay in the first phase of the morning hours is 7.69 min, this delay is 0.36 min in the 4th phase. Such fluctuations are not expected in a dynamically controlled intersection system. This result shows us that due to the extra time given to one phase, other phases are wasted unnecessarily.

As mentioned in the operating principles of the intersection, signalized intersections allocate the cycle time to the connected arms, flows, or phases that it serves in certain orders and makes traffic more efficient for the vehicles. Thanks to the outputs of the queue analysis used to measure the general service level of the intersection, the phase sequences and effective green times calculation model mentioned in Sections 2.5.2 and 2.6 are proposed. It is aimed to complete the existing processes without exceeding the maximum service times determined by the Priority Scheduling method, which is recommended and adapted for the sequential phase model. In the graph below, the proposed phase sequence model is compared with other dynamic models. The proposed model is defined as M1. The results of other working dynamic models are taken from the intersection control devices of the related intersections. The delays in all branches of the three intersections selected from Istanbul and Konya were evaluated. By comparing M1 with the existing adaptive systems, M2 and M3 models, the delay data in 2 h intervals are revealed. Although there is not a significant difference between the proposed model according to dynamic working principles, it can be said that it is quite good according to the results obtained from the

intersection with a fixed operating principle. We can say that the M1 model causes less delay than other models in the heavy traffic time zones of the day.

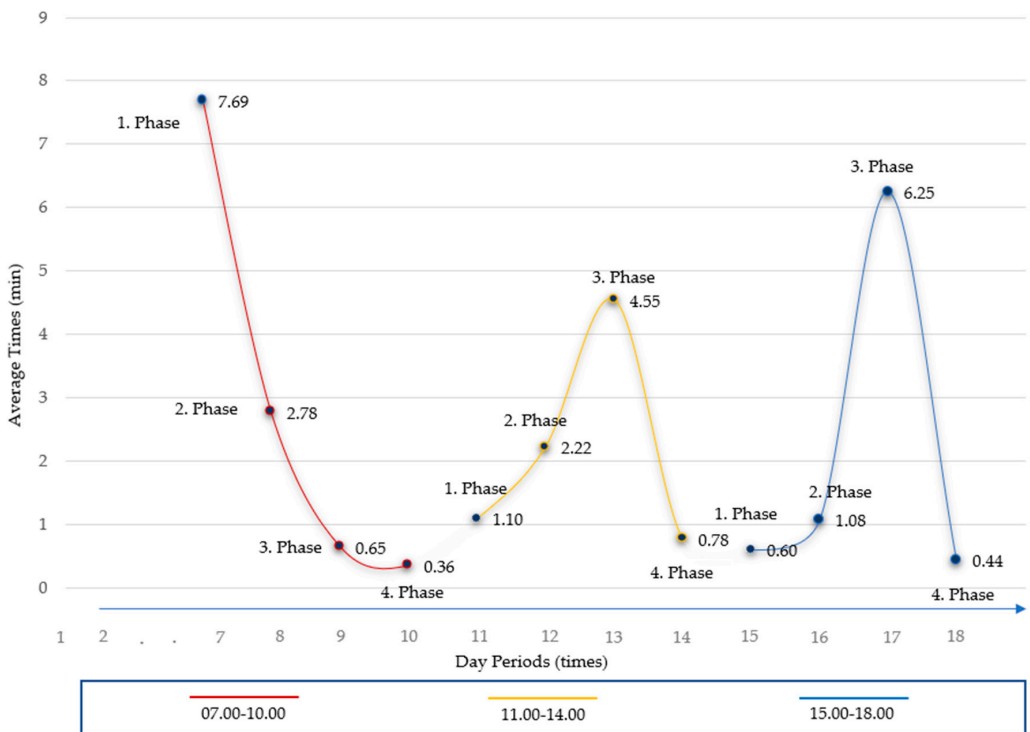

**Figure 11.** The changing average vehicle delays during different periods of a day.

Another important issue in planning and managing intersections is the efficient use of green times. Green periods used efficiently are extremely effective in reducing intensity on intersection roads. Figure 12 shows the change in intensity values of five phases in different periods in a day obtained with improved green times when the proposed model is applied. We can say that the intensities decreased significantly with improved green times. In addition, Figure 13 describes the number of vehicles in the queues before and after model suggested in Section 2.6.

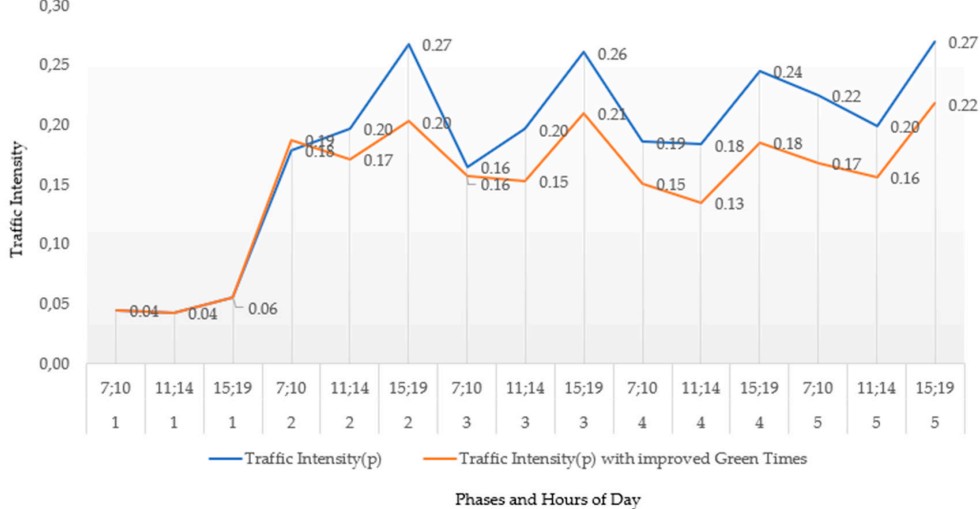

**Figure 12.** Density comparison of phases with improved green times.

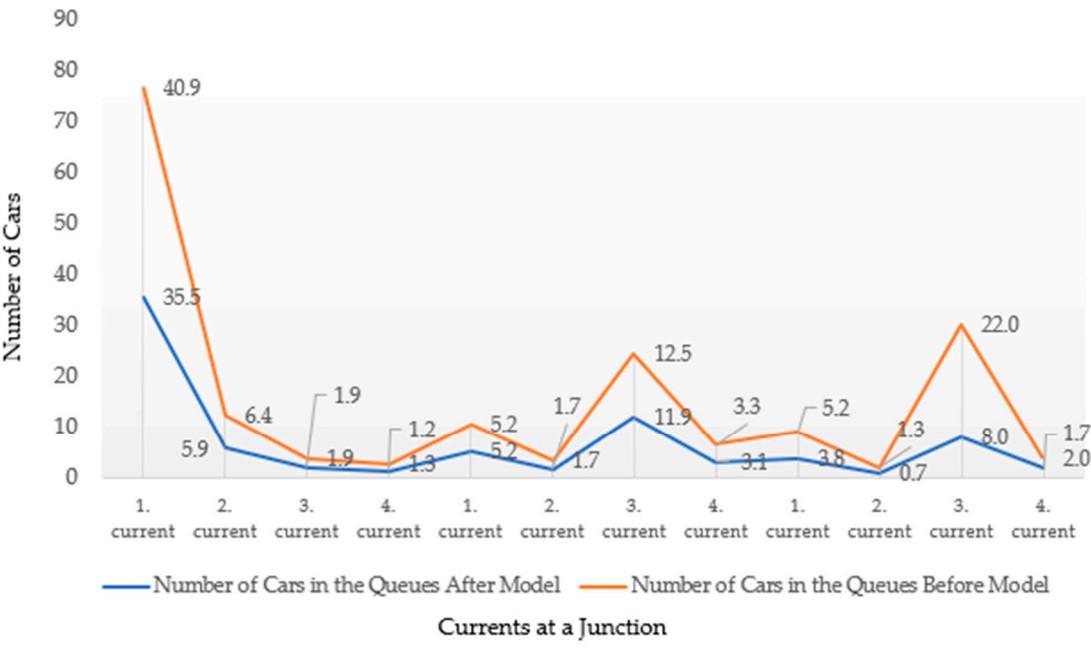

**Figure 13.** The change in the number of vehicles in the queues before and after the proposed model.

Based on the performance values obtained by the methods offered by the queuing theory, it was revealed that the intensity of some intersections was high and the efficiency of the intersections needed improvement. In detailed examinations, it is conspicuous that some currents are intense, especially at certain periods of the day, and that an improvement can be achieved in topics such as saving time, fuel, and energy. Based on the results obtained, a dynamic phase sequence model that allows dynamic managing of the intersection phase sequences is proposed. This model aims to prevent queuing by taking into account the intensity of the intersection arms by assigning priority to these branches in the change of phases. The model was tested in four intersections according to old and new operating principles. When compared to static and dynamic models, as can be seen in Figure 14, the intensity and waiting times, especially at peak hours of the day, were lower.

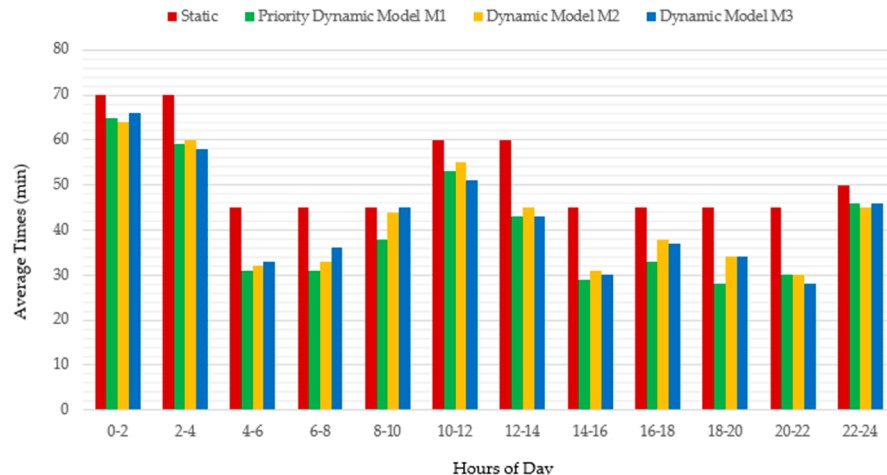

**Figure 14.** Spent time of the vehicles with control model M1 and comparison with another phase control models.

Another issue that was encountered during field studies was the selection of the location and types of sensors to be used. It proved necessary to place cameras inside the intersections and intrusive sensors at certain distances on the arms to obtain the correct lengths of the waiting queues and to make improvements accordingly. Since both sensor

setups were examined during the study, it can be said that both have advantages over the usage situation. For example, cameras are not sufficient for evaluating the flow rate of intersection arms properly. Therefore, sensors placed under the asphalt on roads are very important for the flow analysis of waiting queues. Since the vehicle's motion tracking is an extremely important parameter in the analysis of the phases, camera sensors stand out for the analysis of vehicle types or turning movements of vehicles. In this context, we can easily say that cameras should be used. In summary, it will be beneficial to use both types of sensors in intersection planning and management to obtain efficient results with correct analyses.

## 4. Conclusions

In the studies carried out in Istanbul and Konya provinces, three junctions with three and four connected arms were examined. These intersections are also considered to have both a fixed and dynamic signal pattern. The starting point of this study was the use of queuing models in the approach arms at signalized intersections.

1. The queuing theory, which provides the opportunity to examine queuing problems mathematically, has certain disciplines, as mentioned in Section 2.4 above. In the previous studies that inspired this study, the queuing theory was examined in detail and the examined intersections were modeled depending on their distribution characteristics. Based on the results, it was predicted that improvements were required and, accordingly, new models could be tested in the control of signalized intersections.

2. Two different models are proposed for the use of green durations and the determination of phase sequences. In Figure 11, the average of the spent times experienced by the vehicles in three different intersections are given. It is seen that all three dynamic models manage better than the fixed-time signalized intersection. In addition, it is observed that less time was lost with the M1 model, compared to the other two models, M2 and M3, during rush hours, between 06:00 and 10:00. In addition, it is seen in Figure 12 that the intensity of the flow, which is the most important indicator of queuing and jamming in the intersection connecting arms, is lower with the improvement of green times. However, Figure 13 demonstrates the change in the number of vehicles in the queues before and after the proposed model. As can be seen from the graph, the number of vehicles in the queue is lower after the model. There was a decrease of approximately 21% in the average number of vehicles waiting in the queues during the day.

3. The work principle of the two models M2 and M3 differ from each other due to the sensors used. While loop detectors and wireless sensors are mainly used in M3, vehicle detection systems in M2 work with camera sensors. The main difference here is that sensors positioned on or below asphalt are positioned at certain distances on the junction link arms, while the cameras are positioned close to the intersection center to detect vehicles passing through the intersection endpoint. This fundamental difference in sensor types required performance values to be calculated with different analysis methods. For example, while the queue lengths on the junction approach arms can be calculated more accurately with the loop detectors, this could be calculated using a slightly more complex method with the camera sensors. Therefore, considering the infrastructure cost, using both sensor infrastructures will be more beneficial both in obtaining the number of vehicles more clearly and in distinguishing the vehicle types.

4. There are quite a lot of articles and studies in the literature that stand out with different approaches on this subject. However, it is also observed that most of the research does not go beyond written publication. Because of the lack of practical equivalents of the studies, a clear idea is not given about how it can be applied against different factors that may arise in the field. The prominent uniqueness of this study is the adaptation and testing of the proposed models to traffic control systems in the field. Based on the analysis of the state of the art for control of signal timing and phase

sequences at signalized intersections, there are several factors, such as the location of the intersection, being isolated or coordinated, the capabilities of the sensors, the road capacities, the edges of the connection roads, the presence of irregular entrances other than the connecting roads to the intersection, the traffic use permissions of different types of vehicles during rush hours in the city, the turning density at the exit from the intersection, etc., that affect the performance of the roundabouts in urban areas.

5. Another issue encountered during field studies was the selection of the location and type of the sensors to be used. It was found important to place cameras at intersections and sensors at certain intervals on the arms to obtain the correct lengths of the junction arms and to make improvements accordingly. It was seen that better results can be obtained if both types of sensors are used.

As a result, if the methods and techniques used in traffic problems are carefully considered, a solution will be provided for urban traffic problems. One of the important issues to be emphasized at intersections is the design of the intersections and the infrastructure technologies used. It is important to use improved technology and sensor infrastructures that are no longer costly in all cities. Additionally, according to the results we have drawn, it is obvious that improvements will be made if these suggested models are used in intersection problems.

In future studies, prediction analyses can be made based on the data obtained from a fully equipped intersection and compared with the current results. The same study can be examined on non-isolated intersections and queuing problems in connected intersections can also be investigated. In addition, IoT-based new approaches and concepts, such as V2V, V2I and autonomous vehicles, can be evaluated in the next step of this study. In particular, the V2I concept can lead to different developments in the integration of emergency aid vehicles with intersections or for vehicles to act while approaching the intersection.

**Author Contributions:** Conceptualization, F.G. and S.B.; formal analysis, F.G.; methodology, F.G. and S.B.; supervision, A.H.Z.; writing—original draft, F.G.; writing—review and editing, S.B. and A.H.Z. All authors will be informed about each step of manuscript processing including submission, revision, revision reminder, etc., via emails from our system or assigned Assistant Editor. All authors have read and agreed to the published version of the manuscript.

**Funding:** This research was supported by the Istanbul Commerce University, ISTANBUL.

**Acknowledgments:** The authors would like to thank ISBAK Istanbul IT and Smart City Technologies, ISSD Bilisim Elektronik, the Traffic Management Center of Konya Municipality, Kocaeli Metropolitan Municipality Department of Transportation for supplying data and supporting field studies to conduct this research work.

**Conflicts of Interest:** The authors declare no conflict of interest.

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
