# Peer review of "Smart Cities and Data Analytics for Intelligent Transportation Systems: An Analytical Model for Scheduling Phases and Traffic Lights at Signalized Intersections"

_applsci, doi:10.3390/app11156816_

Round 1

Reviewer 1 Report

This paper is well written and presented. However, I cannot find any interesting contributions.

Therefore, authors should try to improve the presentation of the most remarkable contributions of the paper and their utiity.

Author Response

Dear Editor in Chief,

The required revision has been done according to the reviewer's comments.

We greatly appreciate this valuable reviewer’s comments. To clarify the steps used in this study we did some changes in the manuscript. Different sections have been reviewed according to comments. Especially the conclusion part written in items to make it clearer and to be see the contribution of the study. In terms of English style, the article was proofread with professional support. And to be see the changes in terms of English, an additional file added to the system with the name of “track changes - 1199496”.  Furthermore, to eliminate different considerations were raised by reviewers, these points are explained in detail in this revision and explained below as a response to reviewer comments. The manuscript has been edited in accordance with the comments of the reviewers. Reviewer’s comments have been carefully considered and all revisions recommended by the reviewers have been incorporated. Based on the corrections below, I hope you will re-evaluate and send them back to reviewers. The answers have been given to all reviewers in detail regarding their questions and comments. I hope you and reviewers will be satisfied with this revision and you will reconsider it for publication.

Reviewer 2 Report

The manuscript presents a fundamental theme for managing congestion at traffic light intersections.
It is considered appropriate to include more literature in the introductory section for both the concept of smart cities and the literature on systems architecture.
In particular, we recommend reading the following works 

Garau, C., & Pavan, V. M. (2018). Evaluating urban quality: Indicators and assessment tools for smart sustainable cities. Sustainability, 10(3), 575.

Mannaro, K., Baralla, G., & Garau, C. (2017, March). A goal-oriented framework for analyzing and modeling city dashboards in smart cities. In International conference on Smart and Sustainable Planning for Cities and Regions (pp. 179-195). Springer, Cham.

Pau, G., Campisi, T., Canale, A., Severino, A., Collotta, M., & Tesoriere, G. (2018). Smart pedestrian crossing management at traffic light junctions through a fuzzy-based approach. Future Internet, 10(2), 15.

Ghazal, B., ElKhatib, K., Chahine, K., & Kherfan, M. (2016, April). Smart traffic light control system. In 2016 third international conference on electrical, electronics, computer engineering and their applications (EECEA) (pp. 140-145). IEEE.

the inclusion of a flowchart in the introductory part of the manuscript would exemplify the reading of the research steps carried out 

It is necessary to include more explanation to the case study, defining the road types, the presence of pedestrian crossings and the presence of roadside parking. It is also necessary to define the peak traffic flows characterising the different roads.

The limitations of the present research and future steps should be included in the concluding section.

Since it is written that in the concluding part of this paper, compared to many others, the focus is on a real case study, more photos of the infrastructure equipped with the sensors should be included. 

Author Response

Dear Editor in Chief,

The required revision has been done according to the reviewer's comments.

We greatly appreciate this valuable reviewer’s comments. To clarify the steps used in this study we did some changes in the manuscript. Different sections have been reviewed according to comments. Especially the conclusion part written in items to make it clearer and to be see the contribution of the study. In terms of English style, the article was proofread with professional support. And to see the changes in terms of English, an additional file added to the system with the name “track changes - 1199496”.  Furthermore, to eliminate different considerations were raised by reviewers, these points are explained in detail in this revision and explained below as a response to reviewer comments. The manuscript has been edited in accordance with the comments of the reviewers. Reviewer’s comments have been carefully considered and all revisions recommended by the reviewers have been incorporated. Based on the corrections below, I hope you will re-evaluate and send them back to reviewers. The answers have been given to all reviewers in detail regarding their questions and comments. I hope you and reviewers will be satisfied with this revision and you will reconsider it for publication.

Reviewer 3 Report

Dear Authors,

Congratulations for your work. I am happy to see that all my comments from the previous review (Reviewer 4) were addressed. The overall quality of the paper was consistently improved.

The quality of the figures 9 - 12 can be improved.

Author Response

Dear Editor in Chief,

The required revision has been done according to the reviewer's comments.

We greatly appreciate this valuable reviewer’s comments. To clarify the steps used in this study we did some changes in the manuscript. Different sections have been reviewed according to comments. Especially the conclusion part written in items to make it clearer and to be see the contribution of the study. In terms of English style, the article was proofread with professional support. And to see the changes in terms of English, an additional file added to the system with the name “track changes - 1199496”.  Furthermore, to eliminate different considerations were raised by reviewers, these points are explained in detail in this revision and explained below as a response to reviewer comments. The manuscript has been edited in accordance with the comments of the reviewers. Reviewer’s comments have been carefully considered and all revisions recommended by the reviewers have been incorporated. Based on the corrections below, I hope you will re-evaluate and send them back to reviewers. The answers have been given to all reviewers in detail regarding their questions and comments. I hope you and reviewers will be satisfied with this revision and you will reconsider it for publication.

Sincerely Yours,

Fatih Gunes, PhD

Reviewer 4 Report

  • the manuscript is well structured. the authors dealt with a truly topical theme. I recommend increasing the introductory part by referring to self-driving vehicles. they interact in different scenarios. I recommend implementing the conclusions, they must also refer to future work. increase the bibliography. revise the English grammar. I suggest several bibliographical references:
  • Smart Roads: An Overview of What Future Mobility Will Look Like .S Trubia, A Severino, S Curto, F Arena, G Pau Infrastructures 5 (12), 107
    • Benevolo, C.; Dameri, R.P.; D’Auria, B. Smart Mobility in Smart City Action Taxonomy, ICT Intensity and Public Benefits. In Lecture Notes in Information Systems and Organisation; Springer International Publishing: Cham, Switzerland, 2016. 
    • Ghahari, S.A.; Assi, L.; Carter, K.; Ghotbi, S. The Future of Hydrogen Fueling Systems for Fully Automated Vehicles. In Proceedings of the International Conference on Transportation and Development, Alexandria, Virginia, 9–12 June 2019. 
    • Docherty, I.; Marsden, G.; Anable, J. The governance of smart mobility. Transp. Res. Part A Policy Practic.2018, 115, 114–125. 
  • Decision Tree Method to Analyze the Performance of Lane Support Systems G Pappalardo, S Cafiso, A Di Graziano. Sustainability 13 (2), 846

Author Response

(The authors gave the same response as above.)

Round 2

Reviewer 1 Report

The questions raised in previous comments have been well addressed. The presentation of the paper has been significantly improved. I have no further question.

Reviewer 2 Report

The paper has some typos and grammatical errors.please add image with high resolutions. After this correction it is ready for pubblication 

Reviewer 4 Report

good job!

for me it’s ok !

thank you 

This manuscript is a resubmission of an earlier submission. The following is a list of the peer review reports and author responses from that submission.

Round 1

Reviewer 1 Report

The subject of the article is interesting. However, after reading it thoroughly, the following questions emerged for discussion. 

- Section Anstract, lines 15-22: this text is an introduction rather than a summary of the article; 

- Fig.5: the freehand sketch shown in Figure 5 should be drawn carefully - preferably using the appropriate drawing tools; 

- Fig.6: the symbol "*" is not a multiplication operator used in mathematics;

- Fig.6: the text in the algorithm blocks, especially the variable markings, are poorly readable; 

- Formula (1): the symbol "*" is not a multiplication operator used in mathematics;

- Formula (2): the symbol "*" is not a multiplication operator used in mathematics;

- Table 1, Step 7: the symbol "*" is not a multiplication operator used in mathematics;

- Figure 1: there is no reference to this drawing in the text; 

- Figures 9-12: no OX i OY axle description ;

- Tables 3, 4, 5: variable units are missing in the description of the corresponding columns; 

- Section "5. Conclusions ": the text of this section is an introduction or part of a summary; specific conclusions from the presented analyzes should be rewritten. 

General remarks:

- Introductory content to the issue has been mixed in the text.

For example, the topic of ITS in section 2. Materials and Methods (lines 125-142) would be much better in section 1. Introduction. Similarly, Traffic Management System issues - now included in section 2. Materials and Methods (lines 145-168).

Sections "Literature review" and "State of art" are missing in the context of the presented issue; A very general and insufficient literature review is provided in section 1. Introduction; 

- There is no comparison of the presented approach to the problem and the obtained results with other studies on this issue; 

- There is no explanation of the variables and their symbols on pages: 9-11; 

- Lack of a clearly defined research goal and, above all, a justification of what is new to knowledge, the proposed approach to the problem contributes; 

Reviewer 2 Report

This paper addresses the optimization of traffic flow in road junctions with traffic signals within urban environments. The authors propose a methodology which is applied to some case studies in the city of Istanbul and two other Turkish cities.

In what concerns positive points that can be found in the paper, we can refer the following:

  • The paper is very well written and the graphic quality is good except the handmade drawing in figure 5.
  • The problem addressed in the paper deserves research and new solutions.
  • The paper tries to propose a theoretical model, apply it to several case studies, present results and compare and discuss them.
  • It seams that there is a significant amount of work to produce the paper.

However, the paper is not well organized within each part. The proposals and results presented are not clear enough to be properly evaluated.

First of all, we have to consider the methodology proposed by the authors to address the problem. The authors discuss very briefly scheduling theory, but one can not understand why. If they are targeting to use the results from processor tasks scheduling, a very well studied discipline, or from process scheduling in the industrial domain, they should elaborate much more on the work published in those areas and show how to map the correspondent results on this work. They also discuss just a little bit queuing theory. Again it is not clear how it is applied to their work.

Section 2.6.2, figure 6, section 2.7, related to the model, are not sufficiently detailed to understand and evaluate the model. Here, and everywhere in the paper, the variables are not presently adequately and, often, one can not understand what they represent. Also in the paper there is a complete lack of the definition of the parameters related to traffic engineering. In consequence, it is not possible to analyze the differences from this model to other models available in the literature.

The use cases are not presented in a structured way. They are discussed in a mixed way, jumping from one another. In consequence, it becomes difficult to associate the results to the use cases. One can not also understand why these intersections were chosen to be studied instead of more “canonical” ones like cross shaped. Also, the pictures of the intersections should be more formal having indication of the orientation (where is north?) and of the scale.

The gathering process of values for the data sets is not clear. This sentence, loose in the text, deserves clarification: “The collected data were analyzed in 3-hour, 1-hour, and 15-minute time intervals in 1-day and 10-day periods”. The resulting tables, table 2 and table 3, are confusing, don’t have units. It seams that some parameters, e.g. “Density”, don’t have the same meaning as in traffic engineering.

Section 3 is supposed to discuss the results but is starts to present parameters that are not adequately defined (and that should be defined before). For example, the authors say that a vehicle when “coming from any current enters the intersection arm” gives origin to a t0 instant. After it leaves in a t1 instant. These times are not defined just vaguely described.

The “results” (in “” because they are not at all clear) are shown in tables 4 and 5 that suffer from the same problem of undefined units and parameters. The discussion is also vague and qualitative as it can be shown by the sentence “time losses in phases should not be too high”. What means “too high”?

It is well known that traffic flows depend strongly on the hour of the day, of the day of the week, of the period of the year, of weather conditions and other factors like events. In the results presented there is not a reference of the conditions of the experimental trials.

Graphics have not indication of the variables represented in the axes. Figure 9 is one of the cases and is very odd. In fact, the “average vehicle delays” (not defined) values show two coincidences: the value at nine (9:00) is the same as at eleven (11:00); the value at thirteen (13:00) is the same at sixteen (16:00). How can this happen?

The presentation of results in figures 10, 11 and 12 is unclear. Not only the textual explanation is confusing but also the graphics are not readable. In figure 11 one can not understand what is represented in the YY axis. And in figure 12 in both axes.

In the results discussion there is also a part that comes from nowhere. In fact, the sentence “It was seen that it was important to place cameras inside the intersections and intrusive sensors at certain distances on the arms to obtain the correct tail lengths on the junction arms and to make improvements accordingly.” Does not correspond to anything that has been discussed in the paper.

The paper doesn’t present a section 4.

In section 5 the conclusions are superficial and do not show the paper contributions.

Finally, the references are old and a bit random. For example, the Highway Capacity Manual has already published versions more recent than the one referenced (HCM 2010 published in 2011 and HCM6 published in 2016). It is also recommended to avoid references in Turkish.

In conclusion the paper is not acceptable for publication. A suggestion is to divide it in more than one paper, starting to improve substantially the methodology and providing enough details and an adequate formalization so that the methodology can be understood, thus preparing a publication about it. Whenever this part is accepted than you should try to improve the practical evaluation and results analysis.

Reviewer 3 Report

  1. The abstract is more like a background introduction. Please fill with more of your research work instead of the introductory background.
  2. In figure 1, the author should remove irrelevant information for less distracting the reader. Highlight the parameter that matters such as all the sensors, detectors and so.
  3. In 2.3.1, please align the terms for better explanation. Is semi-constant and simi-fixed same thing? It’s recommended to use a table to introduce and compare different signal plan, which makes the comparison clearer.
  4. When talk about sensors in 2.4, what kind of sensors are they? For counting the vehicle or for measuring the speed?
  5. Please re-plot the figure 5 in a professional way. Add legend and table to clearly describe what kind of sensor are they.
  6. line 343: it’s rho not p.
  7. Figure 6 is really hard to understand. Please describe/explain all the variables in a clear way. Is this model based on constant interarrival time?
  8. The case study is only from one crossing and didn’t have any generalization for other cases, which is not convincing.
  9. Please use quantified data to show the improvement instead just showing figure 11 and figure 12. First, try to find a metrics to evaluate the crossing efficiency. Then evaluate it before and after using the optimization.

Reviewer 4 Report

Please find the review findings in the attached file.

Reviewer 5 Report

This paper addresses the problem of traffic signal control on isolated intersections. A priority scheduling method is proposed and compared to some unexplained existing traffic signal control approaches if I understood the paper correctly. The tackled problem is interesting, but the scientific contribution must be clearly stated, and testing methodology explained and presented precisely and understandable. In this form, this paper is not suitable for publication in a high-impact journal like this one. In the paper revision, pay attention to the following.

When mentioning many reasons or benefits, name a few as an example to be more precise. Make all descriptions easily understandable to the reader.

Explain at the end of the Introduction explicitly what your scientific contribution is compared to existing approaches. You just briefly mention that you propose a new dynamic phase sequence model. Write a few sentences to emphasize its most important features.

Traffic management is a broader term, including also traffic planning, maintenance, user informing, etc. Your paper is focusing only on the control part of traffic management. Make a distinction of this in your paper.

The Priority Scheduling method should be described in more detail if you are using it as starting point.

Explain all the variables you are using in your equations and figures.

Use the term flow, not current, when describing your traffic signal control approach.

Label/explain the axis in your graphs! Give also corresponding measuring units.

Your tables 4 and 5 give a detailed overview of the current performance situation on the use-case intersections. However, when proving your proposed methods' effectiveness, you give only one histogram-based graph without axes labels. When evaluating traffic signal control systems use the Level of Service notations explained in the cited HCM combined with throughput, average queue lengths, Total Time Spent, Total Travel Time, etc., like in https://link.springer.com/chapter/10.1007/978-3-030-18072-0_44, 10.1016/j.trpro.2017.03.084, 10.4236/jtts.2012.23027 or https://ieeexplore.ieee.org/document/9219024.

There are some other comments in the attached PDF. Consider the comments on the used terminology since it is not appropriate for describing traffic control problems.
